# A Double-Blind, Randomized Intervention Study on the Effect of a Whey Protein Concentrate on *E. coli*-Induced Diarrhea in a Human Infection Model

**DOI:** 10.3390/nu14061204

**Published:** 2022-03-12

**Authors:** Laurien H. Ulfman, Joyce E. L. Schloesser, Guus A. M. Kortman, Maartje van den Belt, Elly Lucas-van de Bos, Joris Roggekamp, R. J. Joost van Neerven, Mojtaba Porbahaie, Els van Hoffen, Alwine F. M. Kardinaal

**Affiliations:** 1FrieslandCampina, 3818 LE Amersfoort, The Netherlands; joris.roggekamp@frieslandcampina.com (J.R.); joost.vanneerven@frieslandcampina.com (R.J.J.v.N.); 2NIZO Food Research B.V., 6718 ZB Ede, The Netherlands; joyce.schloesser@nizo.com (J.E.L.S.); guus.kortman@nizo.com (G.A.M.K.); maartje.vandenbelt@wur.nl (M.v.d.B.); elly.lucas@nizo.com (E.L.-v.d.B.); els.vanhoffen@gmail.com (E.v.H.); alwine.kardinaal@nizo.com (A.F.M.K.); 3Cell Biology and Immunology Group, Wageningen University, 6700 AH Wageningen, The Netherlands; mojtaba.porbahaie@wur.nl

**Keywords:** whey protein concentrate, diarrhea, *E. coli* infection, human infection model

## Abstract

Infectious diseases are a major cause of morbidity and mortality worldwide. Nutritional interventions may enhance resistance to infectious diseases or help to reduce clinical symptoms. Here, we investigated whether a whey protein concentrate (WPC) could decrease diarrheagenic *Escherichia coli*-induced changes in reported stool frequency and gastrointestinal complaints in a double-blind, parallel 4-week intervention study. Subjects were randomly assigned to a whey hydrolysate placebo group, a low-dose WPC group or a high-dose WPC group. After 2 weeks of consumption, subjects (*n* = 121) were orally infected with a high dose of live but attenuated diarrheagenic *E. coli* (strain E1392/75-2A; 1E10 colony-forming units). Subjects recorded information on stool consistency and the frequency and severity of symptoms in an online diary. The primary outcome parameters were a change in stool frequency (stools per day) and a change in Gastrointestinal Symptom Rating Scale (GSRS) diarrhea score between the first and second days after infection. Neither dose of the whey protein concentrate in the dietary treatment affected the *E. coli*-induced increase in stool frequency or GSRS diarrhea score compared to placebo treatment. The composition of the microbiota shifted between the start of the study and after two weeks of consumption of the products, but no differences between the intervention groups were observed, possibly due to dietary guidelines that subjects had to adhere to during the study. In conclusion, consumption of the whey protein concentrate by healthy adults did not reduce diarrhea scores in an *E. coli* infection model compared to a whey hydrolysate placebo control.

## 1. Introduction

Diarrheal disease is a common cause of illness and death worldwide, especially in children under 5 years of age [1]. Pathogenic *E. coli* strains are among the most common causes of diarrheal diseases in Asia and Africa [2]. Access to safe water, sanitation, vaccination programs and nutrition are important factors in the prevention of diarrheal disease [3,4].

Dairy ingredients have been shown to affect infectious diseases in preclinical studies as well as in in vivo intervention trials [5,6]. Both native whey components, such as bovine immunoglobulins (reviewed in [5]) and lactoferrin (reviewed in [7]), and milk fat globule membrane (MFGM) components (reviewed in [6,8]), such as phospholipids and MFGM proteins, may contribute to these effects. With respect to the MFGM type of products, some [9,10,11,12] but not all [13,14] studies have shown a reduced incidence of infectious outcomes of interventions for infants [9,10,11,14] and children [13]. Three [9,11,12] of the four studies that found an effect of the intervention should be interpreted with caution since infection was not defined as the primary outcome of the study. One study was hampered by a low incidence of diarrheal disease [14]. Another study did find that febrile episodes decreased, suggesting an immune-modulating effect [13].

Double-blind, randomized intervention field trials are the gold standard to study the effects of nutritional interventions on the prevention of diarrheal disease. However, these studies are very time-consuming and costly and therefore not always the first choice to test a new food ingredient. Human pathogenic infection challenge models are therefore a very relevant alternative as the first step in testing whether a nutritional intervention can affect the course of a natural infection.

The human infection challenge model used was a diarrheagenic *E. coli* strain challenge developed at NIZO (The Netherlands) [15]. The *E. coli* used is a well-characterized, antibiotic-susceptible organism that has been associated with mild diarrhea and gastrointestinal symptoms (severity and duration). Indeed, previous intervention studies using the diarrheagenic *E. coli* strain showed transient induction of symptoms, including increased stool frequency, mild diarrhea, mild abdominal pain and bloating [16,17,18]. The diarrheagenic *E. coli* strain E1392/75-2A is a spontaneous mutant with the deletion of the genes encoding heat-labile (LT) and heat-stable (ST) toxins and therefore cannot produce any toxins. However, it expresses colonization factor antigen II (CFA/II) and provides 75% protection against challenge with an LT, ST, CFA/II strain [19,20]. This model has been successfully used to test the effect of nutritional interventions, including a phospholipid-rich product [21].

As described above, whey proteins and MFGM components, such as specific proteins and phospholipids, have been associated with anti-infection and immune-related outcomes. The whey protein concentrate tested in this study contained these components. The main objective of the current study was to test the effect of two concentrations of a whey protein concentrate on *E. coli*-induced diarrheal disease in otherwise healthy adults. The effects were measured by the number of stools per day and diarrhea score between days 2 and 3 after infection as compared to the control group.

## 2. Materials and Methods

### 2.1. Trial Design

The trial had a parallel design with 3 treatment arms and a 4-week intervention duration, of which 2 weeks were before the *E. coli* challenge and 2 weeks were after the *E. coli* challenge (22 October 2020–4 December 2020). The set-up was comparable to a previous study in which a phospholipid-rich dairy product was tested for its reducing effect on *E. coli*-induced stool frequency and gastrointestinal complaints in adults [21]. Participants visited the study site (NIZO, Ede, The Netherlands) 3 times, once for information on the trial, once for baseline measurements and once for *E. coli* challenge. All other interactions with participants were via online tools and courier services for picking up samples/materials. The study schedule is available in Appendix A.

### 2.2. Participants, Ethics and Trial Registration

This human intervention study was approved by the Medical Ethics Committee (METC) of Brabant, Tilburg, The Netherlands (July 2019), and registered as Protocol NL66645.028.18. In addition, the study was registered at the Netherlands Trial Register as NTR7613 NTR (trialregister.nl, 9 November 2018). Healthy male subjects between 18 and 55 years with a BMI (Body Mass Index) between 18.5 and 30 kg/m^2^ were included in the study. For a complete in- and exclusion criteria list, see Appendix A. Subjects who were willing to participate were asked to sign an informed consent form and were checked for eligibility by the Principal and Medical Investigator. Participants (149) provided informed consent, of which 121 were eligible for the study and of which 116 completed the study. For details on screening failures (*n* = 28) and dropouts (*n* = 5), see the subject flow diagram in Figure 1.

### 2.3. Randomization and Stratification

Subjects were stratified by an independent NIZO scientist according to age and BMI and randomly assigned to one of three treatment groups: placebo, WPC low dose and WPC high dose. Stratified randomization of subjects to the treatment group was performed using the Research Manager software version 5.40 (Deventer, The Netherlands).

Study products were supplied by FrieslandCampina, and each of the 3 products received 3 unique codes. Blinding and labeling of the investigational product (IP) was performed manually by the independent NIZO scientist by placing a label with the right subject code on the assigned study product, which covered the unique code.

All researchers on the NIZO project team, with the exception of the independent NIZO scientist, were kept blind to the assignment of treatment, as were the study subjects.

The randomization code was kept by the independent scientist. The investigators and participating subjects were blinded until after the blind data review. After sending the approved master data file to the statistician, the key to the treatment allocation was sent to the statistician by the independent NIZO scientist.

The randomization code of the study was broken after (i) all laboratory reports related to primary and secondary outcomes were authorized by the Principal Investigator, (ii) the data master file was documented as meeting the cleaning and approval requirements of the Principal Investigator, and (iii) the statistical analysis plan was finalized and approved by the Principal Investigator.

### 2.4. Medical Guidelines

Use of medication known to affect GI functionality, including non-steroidal anti-inflammatory drugs, acid suppression medication and antimotility agents, was prohibited three days before, during and four days after the diarrheagenic *E. coli* challenge (day 11 until day 18). Antibiotics were prohibited during the run-in period and throughout the whole study. Paracetamol (up to 2 g/day) was allowed as rescue medication during these days. Subjects received a list of prohibited and permissible medication. However, if medication was prescribed by the general practitioner or study physician, subjects were allowed to take these medications but needed to record intake (dose and frequency).

### 2.5. Dietary Guidelines and Calcium Restriction

Subjects were instructed to maintain their habitual diet, except for their dairy intake. Dairy has a high calcium content and contributes significantly to total daily calcium intake in the Dutch diet [22]. From previous studies, it is known that calcium can significantly reduce the gastrointestinal symptoms induced by the *E. coli* strain [17]. Therefore, subjects were asked to omit dairy products. It was estimated that this restriction in dairy intake would lower the subject’s daily calcium intake without the treatment to 400–500 mg. Including the additional calcium intake through the intervention products (430–460 mg/day), overall calcium intake was in the range of the recommended daily calcium intake of 1000 mg in The Netherlands.

### 2.6. Probiotics and Prebiotics/Fibers

Subjects were instructed to avoid probiotics, non-dairy yogurts with active cultures of probiotics, probiotic-based supplements, products with added prebiotics/fiber and prebiotic/fiber supplements starting from the run-in period and lasting for the duration of the whole study due to potential effects on the duration of gut infection-induced diarrhea.

### 2.7. Dietary Intervention Products

#### Nutritional Values

The investigational products (IPs) were partially hydrolyzed whey protein (as placebo) and a whey protein concentrate (WPC) as the test product. All products were produced by FrieslandCampina, The Netherlands. All products were mixed with 18% maltodextrin to increase the solubility of the powder. The high dose of the WPC was pure WPC, and the low WPC dose was a mixture of WPC and the placebo product at a ratio of 0.9:1. All products were manufactured under the scope of a certified COKZ Z0949 approved food safety management system. Table 1 shows the nutritional composition of the three IPs in % weight. Since calcium is a potential confounder in the study, the calcium levels were equalized in all 3 products. Subjects were asked to slowly dissolve the content of a sachet (28 g) in 180 mL of stirring water using a vortex provided to them. Subjects were asked to consume the product twice a day, during breakfast and during an evening meal. The control product had a bitter taste that was masked with instant coffee or syrup. To prevent deblinding, all participants were asked to use either syrup or instant coffee powder with or without sweetener. Usages of these taste options were equally distributed over the 3 groups. Questions from subjects about the study product were asked to an independent staff member to keep the study team blind at all stages.

### 2.8. Compliance with Dietary Guidelines and Supplement Intake

Participants were requested to record daily IP intake and indicate whether they complied with the dietary guidelines in their online diary. IP compliance was cross-checked with the returned sachets of IP.

### 2.9. Diarrheagenic E. coli Challenge

For logistic reasons, the subjects were randomly split into 2 cohorts, which followed an identical study schedule, with a 1-week delay for the second cohort.

After a run-in period of 13 days, on day 13, a standardized meal was provided to the subjects. On study day 14, after an overnight fast, a blood sample was drawn. Under the supervision of the project team, subjects drank a NaHCO3 solution (100 mL of 2% NaHCO3) to neutralize gastric acid. After 5 min, they drank a fruit syrup drink (100 mL) containing the attenuated diarrheagenic *E. coli* strain E1392/75-2A (1E10 CFU). Subjects went home but were not allowed to drink or eat for 1 h, after which they consumed the breakfast dose of the investigational product.

### 2.10. Reported Stool Consistency, Stool Frequency and Gastrointestinal Symptoms

Subjects had to report information on stool frequency, stool consistency based on the Bristol Stool Scale (BSS) [23] and medication intake. Additionally, subjects had to record the frequency and severity of symptoms related to reflux, abdominal pain, indigestion, diarrhea and constipation by using the validated Gastrointestinal Symptom Rating Scale (GSRS) [24]. The GSRS is a disease-specific instrument of 15 items combined into 5 symptom clusters. The GSRS has a 7-point graded Likert-type scale, where 1 represents the absence of troublesome symptoms and 7 represents very troublesome symptoms. The online data management system ‘De Research Manager’ was used to record all subject information.

### 2.11. Fecal Sample Collection

Fecal samples were collected one day before inoculation (day 12 or 13), on day 14 (inoculation day), for 4 days after the inoculation day (day 15 until day 18) and on the last study day (day 27 or 28). Subjects were asked to collect all 24 h stool samples in collection bags and freeze them on-site using a provided mini freezer. The frozen samples were transported to NIZO, where they were sorted, homogenized, aliquoted and stored at −20 °C until later analysis.

Microbiota and SCFA were analyzed in samples from subjects who completed the entire study, had >90% compliance with study product intake requirements, had 100% compliance with study product intake requirements on days 11, 12, and 13 and collected a stool sample at baseline and on day 12/13. This resulted in *n* = 33 samples from the placebo group, *n* = 30 from the WPC low group and *n* = 27 from the WPC high group. Of these 90 subjects, the fecal samples at the start (before consumption of product, *t* = 1) and on day 12 or 13 (before inoculation, *t* = 2) were analyzed for microbiota and SCFA to investigate the effect of 2 weeks of consumption of the IP.

### 2.12. Microbiota Composition Profiling by 16S rRNA Gene Sequencing

DNA isolation, including vigorous bead-beating steps, was performed as described previously [25]. Barcoded amplicons from the V3–V4 region of 16S rRNA genes were generated using 2-step PCR and according to previously described methods [25]. For the second PCR in combination with sample-specific barcoded primers, purified PCR products were shipped to BaseClear BV (Leiden, The Netherlands). PCR products were checked on a Bioanalyzer (Agilent, Santa Clara, USA) and quantified. This was followed by multiplexing, clustering and sequencing on an Illumina MiSeq (San Diego, USA) with the paired-end (2×) 300 bp protocol and indexing. FASTQ read sequence files were generated using bcl2fastq2 version 2.18. Initial quality assessment was based on data passing Illumina Chastity filtering. From the raw sequencing data, the sequence reads of insufficient quality (only “passing filter” reads were selected) were discarded, and reads containing adaptor sequences or PhiX control were removed with an in-house filtering protocol. On the remaining reads, quality assessment was performed using the FASTQC tool version 0.11.5 (Babraham Bioinformatics, UK; https://www.bioinformatics.babraham.ac.uk/projects/fastqc/ (accessed on 23 February 2022))

Sequences of the 16S rRNA gene were analyzed using a workflow based on Qiime 1.8 [26]. On average, 45,862 (range 13,179–57,858) 16S rRNA gene sequences per sample were analyzed. We performed operational taxonomic unit (OTU) clustering (open reference), taxonomic assignment and reference alignment with the pick_open_reference_otus.py workflow script of Qiime, using uclust as the clustering method (97% identity) and GreenGenes v13.8 as the reference database for taxonomic assignment. Reference-based chimera removal was performed with Uchime (Drive5 Bioinformatics software and services; http://drive5.com/usearch/manual/uchime_algo.html (accessed on 23 February 2022)) [27]. The RDP classifier version 2.2 was used for taxonomic classification [28].

### 2.13. Fecal SCFA Analysis

For organic acids (lactate, acetate, propionate, butyrate, isobutyrate, valerate and isovalerate), fecal samples were prepared according to a modified and previously described method (see also the Appendix A and Methods) [29].

### 2.14. Data and Statistical Analysis

#### Sample Size Calculation

Sample size calculation was based upon 2-sided statistical testing, unpaired analysis, α = 0.05 (chance of type I error) and ß = 0.20 (chance on type II error) and took the 3 study arms into account with the expected dose–response of the 2 WPC arms. Based on reported stool frequency (number of stools per day) and diarrhea complaints (Gastrointestinal Symptom Rating Scale, diarrhea domain) in an earlier study [21] and taking into account potential dropouts, pre-exposure of subjects to *E. coli* (1–2 per study based on experience in the past) and a possible effect depending on whether the subject was in the first or the second logistic group, the calculated sample size was 40 subjects per arm. In total, this resulted in *n* = 120 subjects for the study, of which 121 were realized (see below).

### 2.15. Statistical Analysis of Primary Outcomes

Intention-to-treat (ITT) and per protocol (PP) analyses were performed for all outcomes and for all randomized subjects who received the challenge and for whom results for the primary outcomes were available. ITT analysis included all consented and randomized subjects, having (reportedly) consumed the study product at least once; PP analysis excluded subjects with protocol violations or major protocol deviations. Criteria for subjects to be excluded from the PP analysis were as follows:-Subjects who did not fulfill the in- and exclusion criteria;-Subjects who reported protocol violations or major protocol deviations;-Subjects for whom IP compliance was <80%;-Subjects for whom >50% of the data points were missing.

After inclusion, 5 subjects dropped out of the study. One subject (193/131) had to be excluded from the study before the 2nd cohort started with the trial. This subject was replaced in the 2nd cohort. The other 4 subjects dropped out before the infection day. Therefore, 116 healthy males completed the study (see also Figure 1).

### 2.16. Outcome Analysis

Primary outcomes were a change in the number of stools per day, d15-16, and a change in GSRS score, domain diarrhea, d15-16. These are continuous variables and were analyzed using mixed model analysis, with study treatment and baseline variable on day 15 as fixed effects and age, BMI and logistical groups as covariates. Correction for multiple testing was carried out by correcting the threshold using the Hochberg procedure because 2 primary outcomes were defined [30]. If both primary parameters had a *p*-value below the threshold of 0.05, both null hypotheses were rejected. If one *p*-value was above 0.05, the other one must be lower than 0.025 to reject the null hypothesis and thus accept a significant effect of the intervention.

Secondary outcomes were (a) change in stool consistency measured by BSS score (1–7) between days 15 and 16 using a chi-square test, (b) percentage change in stool frequency on d16 vs. d15 using a mixed model with study dose and baseline variable on day 15 as fixed effects and age, BMI and logistical groups as covariates, (c) stool consistency measured as mean BSS score per day and maximum BSS score per day, where change and percentage change for both parameters were analyzed using the same mixed model as that used for the primary parameters, (d) kinetics of gastrointestinal symptoms (GSRS), measured as sum total GSRS score, sum domain diarrhea score and sum domain abdominal pain score. Changes and percentage changes for all parameters from days 14 to 18 were analyzed using mixed model repeated measures (MMRM) with dose and time points as the fixed factors and age, BMI and logistical group as covariates, (e) maximum GSRS score, measured by sum total GSRS score, sum domain diarrhea score and sum domain abdominal pain score from days 14 to 18, analyzed using ANCOVA with dose as the factor and age, BMI and logistical group as covariates.

SCFAs were analyzed as an exploratory outcome using the same mixed model as for the primary parameters. Analyses were performed according to a statistical analysis plan, and all analyses were performed using Stata, version 12.

### 2.17. Microbiota Analysis

Between-treatment group differences in alpha-diversity (Faith’s phylogenetic diversity, Shannon index and Richness) were assessed by the non-parametric Kruskal–Wallis test with Dunn’s post hoc test. Alpha-diversity differences between t1 and t2 of the combined treatment groups were assessed by the Wilcoxon signed-rank test. In the bivariate exploratory analysis of all taxa, the Mann–Whitney U test with FDR (Benjamini–Hochberg) correction for multiple testing was applied to assess differences between two treatment groups. Taxon relative abundance differences between t1 and t2 in the combined treatment groups were assessed by the Wilcoxon signed-rank test, followed by FDR correction. For longitudinal analysis of the change in taxon relative abundance over time, 2log ratios were calculated, in which the relative abundance of a taxon at endpoint was divided by the relative abundance of the same taxon at baseline. Ratios were compared between groups by Mann–Whitney U tests with FDR correction for multiple testing.

Redundancy analyses (RDAs) of the gut microbiota composition, as assessed by 16S rRNA gene sequencing, were performed in Canoco version 5.11 (Canoco 5, http://www.canoco5.com (accessed on 23 February 2022)) using default settings of the analysis type “Constrained” [31]. Relative abundance values of OTUs were used as response data and metadata as the explanatory variable. For visualization purposes, genera (and not OTUs) were plotted as supplementary variables. Longitudinal effects of the intervention were assessed by calculating 2log ratios in which the relative abundance of an OTU or genus at endpoint was divided by the relative abundance of the same OTU or genus at baseline. These ratios were used as response variables in RDAs and were weighted based on the average relative abundance of each OTU in all subjects. RDA calculates *p-*values by permutating (Monte Carlo) the sample status. Partial RDA was employed to account for covariance attributable to age and BMI; these were first fitted in the regression modeling and then partialled out (removed) from the ordination, as described in the Canoco 5 manual [31].

## 3. Results

### 3.1. Baseline Characteristics and Compliance with Diet and Investigational Products

In total, 121 healthy male human subjects who fulfilled all inclusion criteria and none of the exclusion criteria were included in the study (see Figure 1 for the flow diagram of study participants). Five subjects dropped out before the infection day, resulting in 116 subjects who completed the study.

Based on online diaries and the checking of returned empty sachets at the study location, in total, 5 subjects had compliance lower than 80% and were therefore excluded from the PP population. All subjects had a compliance of more than 80% with the calcium restrictions. One subject used medication in the restricted period and was also excluded from the PP population. The baseline characteristics of the PP population (*n* = 110) are depicted in Table 2. The results presented here are for the PP population.

### 3.2. Primary Outcomes

The main objective of the current study was to test the effect of a whey protein concentrate on *E. coli*-induced diarrheal disease in otherwise healthy adults, as measured by a faster decline in stool frequency and diarrhea symptoms between day 15 (one day after infection) and day 16 (two days after infection) compared to placebo. Figure 2 shows the stool frequency (Figure 2A) and diarrhea score (Figure 2B) of the three treatment groups during the complete trial. On day 14, the *E. coli* challenge was performed, which resulted in an increase in symptoms on day 15 in all groups. On day 16, the symptoms declined as expected.

There was an overall significant effect of treatment dose on stool frequency (*p* < 0.01), as shown in Table 3. Indeed, the difference between the high-dose and placebo groups suggested a somewhat delayed recovery in the high-dose group, but this difference was not significant (*p* > 0.025) after correction for multiple testing [30]. Post hoc analysis of primary outcomes taking only the high responders (>1 stool on day 15) into account also did not show significant differences between the groups, excluding a diluting effect of the non-responders on the primary outcomes (data not shown).

### 3.3. Secondary Outcome

Secondary study outcomes were tested in three areas of *E. coli*-induced infection effects: stool frequency (Table 3), gut comfort complaints (Table 4) and stool consistency (Table 5). No significant effect was found in the PP analysis for any of these secondary outcomes. In the ITT analysis, the % change in stool frequency between days 15 and 16 was statistically significant (*p* = 0.035) between the intervention groups, as was the percentage change in GSRS diarrhea score (*p* = 0.035). However, a sensitivity analysis of these parameters did not confirm these results (*p* = 0.112 and *p* = 0.249, respectively).

### 3.4. Post Hoc Analysis

Post hoc analyses were performed in two areas of *E. coli*-induced infection effects: stool frequency and GSRS diarrhea. Stool frequency and GSRS diarrhea scores on day 15 (the peak of the infection) were compared between the groups. Furthermore, the areas under the curve for both stool frequency and GSRS diarrhea scores were compared between the groups (Table 3 and Table 4, indicated with ^PH^). Again, no significant differences were found, nor were they found for the other subdomain of the GSRS, abdominal pain (data not shown).

### 3.5. Microbiota Results

Exploratory analysis of microbiota composition was performed to test the hypothesis that a whey protein concentrate modulates the microbiota differently compared to the whey hydrolysate placebo. Microbiota composition at the start of the intervention was compared with microbiota obtained after 2 weeks (day 12/13 of the study) of consuming the intervention products but before the *E. coli* infection. In line with the primary outcomes, microbiota diversity (Faith’s phylogenetic diversity, Shannon index and Richness) was not different between the intervention groups at either time point. From baseline to day 12/13, Shannon diversity significantly increased (*p* = 0.0176) in the study population (combined treatment groups), but Faith’s phylogenetic diversity and Richness were not different (Appendix A). Cross-sectional analysis by redundancy analysis (RDA) showed no difference between the microbiota composition of the WPC low, WPC high and placebo groups at d12/13 (variation explained by treatment was 0.1%, *p* = 0.36). However, time was a significant effect across all groups (variation explained by time point was 2.5%, *p* = 0.002), with a similar effect in all groups (Figure 3). Baseline (t1) was associated with higher relative abundances of, e.g., *Streptococcus* and *Lactococcus*, while d12/13 (t2) was associated with, e.g., *Parabacteroides* and *Odoribacter.* Bivariate analysis of all taxa across all groups between baseline and d12/13 showed, e.g., lower *Streptococcus* and *Lactococcus* (*p* = 0.0015 and *p* = 0.0009, respectively) and, e.g., higher *Odoribacter* and *Bacteroides* (*p* = 0.0005 and *p* = 0.0144, respectively) at d12/13 (Appendix A). However, RDA of change over time (ratios) confirmed that the microbiota change was not different between treatment groups (variation explained 0.4%, *p* = 0.21). Additionally, no association of microbiota composition with complaint scores was found (data not shown). In line with the lack of difference in microbiota composition between the intervention groups, there was also no observed difference in fecal SCFA composition (Appendix A).

### 3.6. Adverse Events

Overall, study execution was safe, and adverse events did not deviate from expectations. During the study, 124 adverse events (AEs) were reported, of which 49 were probably related to the *E. coli* inoculation, 22 were probably related to the investigational products and 53 were probably not related to the study. No serious adverse events were reported during the study.

## 4. Discussion

Dairy products and ingredients are of interest as a source of nutrition and can also be a source of immune-active components [5,8,32,33]. Indeed, many bovine proteins and complex lipids are highly similar to their human counterparts [34,35] and show functional responses in vitro [5,36,37,38,39] and in vivo [5,6,7,8,9,10,13,40,41].

Whey protein concentrates have been studied by multiple research groups in clinical intervention trials [9,10,12,13,42], especially in infants because of the relatively high demand for immune support at that vulnerable age. Most of these whey protein concentrates contain proteins and lipids associated with the milk fat globule membrane [43]. Here, we used an adult infection model using *E. coli* to study the effect of a whey protein concentrate.

In the present study, dietary treatment with WPC did not lead to a faster recovery from *E. coli*-induced diarrhea compared to a control hydrolyzed-whey-containing product, as concluded from the primary outcomes (ITT and PP). Although a significant effect was observed in secondary outcomes, namely, % change in stool frequency and % change in GSRS diarrhea score in the ITT population, suggesting a slower recovery, this was not robust, as concluded from the sensitivity analyses. Additionally, for the PP population, in which all subjects were compliant with study product intake requirements, no significant effects were found. The primary outcomes were selected based on the results obtained in an earlier study with a phospholipid-rich dairy product [21]. In that study, a significant effect was found on the secondary outcomes stool frequency and gastrointestinal symptoms in the days immediately after the infection.

The reason for not finding an effect in the present study is not known but may in part be explained by several factors. First is the type of protein that the products contain. In the current study, the WPC contained >95% whey proteins, as did the placebo, although hydrolyzed in the latter. The protein fraction of MFGM has been shown to have anti-*E. coli* effects in vitro [38], but no effect of the whey hydrolysate on the study outcomes was expected. However, it is possible that some of the peptides or amino acids present in the hydrolysate could have antimicrobial or immune-modulating activity [44]. It can therefore not be fully excluded that the whey hydrolysate had some effect on the outcomes, thereby potentially masking an effect of WPC. However, a recently published study using the same dose of *E. coli* showed a very comparable degree of *E. coli*-induced diarrhea responses in a non-treated population [15], making it less likely that the placebo in the current study suppressed diarrhea responses.

The phospholipid component, which has been shown to have an antibacterial effect, reviewed in [45], was highly comparable in composition (data not shown) and dose (WPC high) between the current study and previously published data [21] and thus cannot explain the observed differences.

The intake of protein in the current study was 30 g. One hypothesis is that 30 g of high-quality nutritious whey protein effectively supported the immune system in the WPC high, WPC low (which is a mixture of WPC and hydrolysate) and hydrolysate groups. Indeed, an increased intake in daily protein consumption to meet the demand during acute infection has been estimated [46,47], and immune function is dependent on available amino acids [48]. Therefore, it is possible that both sources of high-quality whey proteins used in our study were sufficient for a fast recovery from the infection challenge in healthy adults. Arguments against this are that this population was already a well-nourished population, and furthermore, the severity of the *E. coli*-induced diarrhea in the current study was comparable to that in a previous study [15]. A limitation of the study is that only one dose was tested. Whether a higher dose would have had an effect remains to be established.

A potential confounder in the current study using the E1392/75-2A *E. coli* challenge model is dairy-derived calcium phosphate, which was previously shown to confer protection against gastrointestinal infection in this model [17]. Comparison of a dose of 1100 mg versus 60 mg resulting in ~1500 mg vs. ~400 mg daily calcium intake between the two groups resulted in a reduction in *E. coli-*induced infection in this model. In the current study, 430–460 mg calcium per day was obtained from the study products, and an additional ~400 mg was obtained from the calcium-restricted diet. As a result, all participants consumed around 900 mg of calcium daily. Since a dose–response study has not shown at what concentration protection by calcium is apparent, it cannot be excluded that calcium in the current study had a dampening effect on the diarrhea symptoms in all subjects. Again, a recent study using the same model in which no investigational product was used but where subjects did comply with a low-calcium diet (<500 mg calcium in feces/day) [15] showed that *E. coli* induced a very similar symptom pattern compared to the current study, arguing against a suppressed diarrhea effect because of higher total calcium. Indeed, post hoc analysis showed that even in the subgroup analysis of the high responders, no treatment effect was found.

Consumption of WPC had no differential effect on fecal microbiota composition and diversity between the low-dose, high-dose and placebo groups after two weeks of intervention. This indicates that the whey proteins and phospholipids in the WPC do not modulate gut microbiota composition differently when compared to the control, hydrolyzed whey. Although effects on the primary readouts and on the fecal microbiota may be independent, the comparable gut microbiota composition between the intervention groups is consistent with not finding an effect on the primary readouts. Most striking was the change in microbiota composition and diversity/evenness (Shannon index) over time, which was observed for all intervention groups. The most likely explanation for this change over time is the change in diet in accordance with the dietary guidelines, as all subjects were instructed to adhere to a calcium-restricted diet, specifically omitting dairy products. However, the specific dietary drivers for the microbial compositional change could not be identified. Calcium intake may affect the gut microbiota, but we did not specifically determine the actual calcium intake of the subjects. The restriction of dairy products and supplementation of the intervention products may have influenced the casein/whey ratio, but this ratio was previously shown to be an unlikely driver of microbiota changes [49]. Interestingly, a high-dairy diet has previously been found to increase *Streptococcus*, *Lactococcus* and *Leuconostoc,* while *Faecalibacterium* decreased, but Shannon diversity was not different [50]. In the current study, *Streptococcus* and *Lactococcus* were associated with baseline (before dairy restriction), while *Faecalibacterium* was one of the taxa associated with the two-week time point (after dairy restriction), and therefore, the change in microbiota may be the result of dairy restriction.

In this study, we used the *E. coli* infection challenge model to study the effects of the intervention on infection and diarrhea as a model to evaluate whether the WPC might offer protection against infection in infants. A limitation of this study is that this type of inoculation can only be performed in healthy adults with a well-functioning immune system [51]. This population does not represent a vulnerable infant population with a high disease burden.

Field trials in infants have shown that whey protein concentrates, which contain milk fat globule membrane components, may have a beneficial effect on infection outcomes [10,13]. Additionally, multiple studies have shown beneficial effects of polar lipids and specific proteins present in the MFGM fraction on diverse infectious microbes in preclinical [40,52] and in vitro studies [37,38,39,53,54,55,56]. In the current study, a short nutritional intervention with a WPC showed no effect on acute gastrointestinal infection symptoms in adults. Whether the current WPC offers protection against infection symptoms in an infant population remains to be established.

## 5. Conclusions

Short-term consumption of a whey protein concentrate by healthy adults (a) did not reduce diarrhea scores in an *E. coli* infection model and (b) did not modify the composition of the intestinal microbiota. More research is needed to exploit the potential effects of whey protein concentrates on infection outcomes in vivo.

## Figures and Tables

**Figure 1 nutrients-14-01204-f001:**
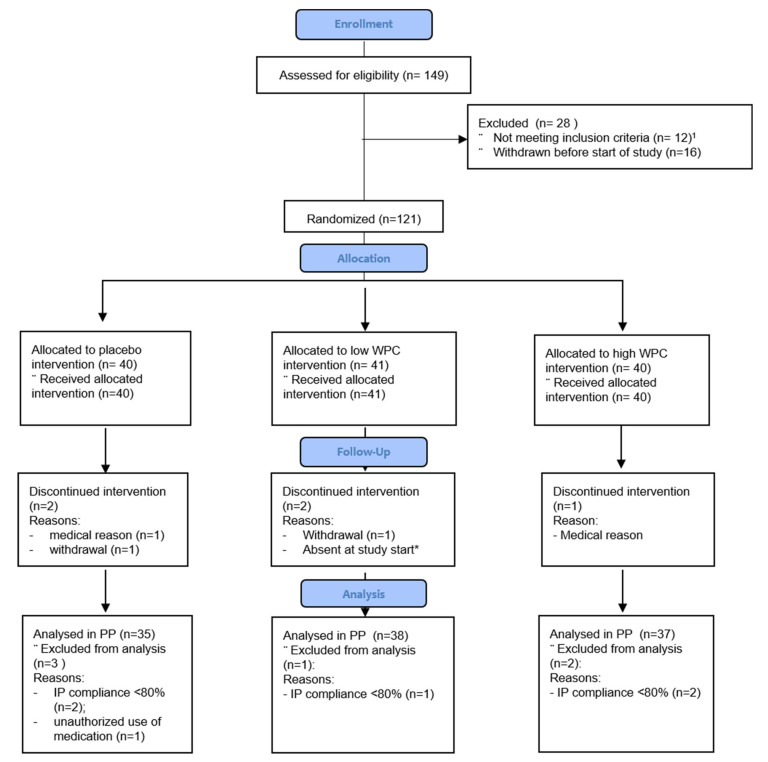
Flow diagram of the GIGA study. ^1^ Reasons for not meeting inclusion criteria: participation in another study (*n* = 1), frequency of defecation (*n* = 3), travel diarrhea (*n* = 4), prior participation in ETEC study (*n* = 1), BMI (*n* = 1), diagnoses IBS (*n* = 1) and planned operation during infection week (*n* = 1). * This person was replaced by another person in cohort 2 that started a week later. Discontinued interventions in total were *n* = 5 (dropouts).

**Figure 2 nutrients-14-01204-f002:**
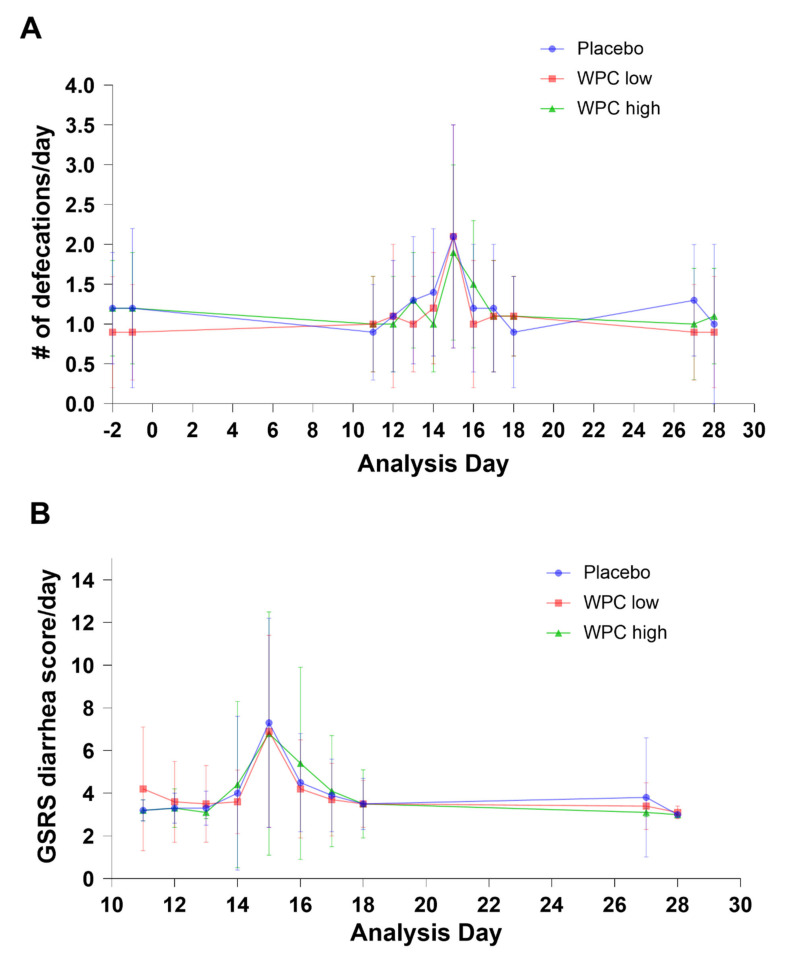
Response curves for stool frequency (mean # of defecations per day +/− SD) (**A**) and GSRS (mean domain score diarrhea per day +/− SD) (**B**) for the three treatment groups in the per protocol population. No statistically significant differences were found between placebo (circles, blue), WPC low (squares, red) and WPC high (triangles, green) using mixed model statistical analysis.

**Figure 3 nutrients-14-01204-f003:**
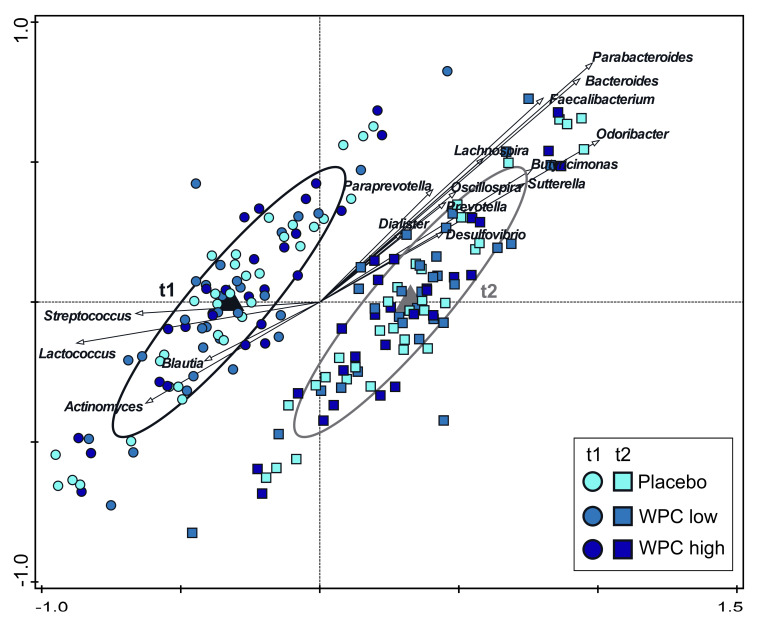
Redundancy analysis (RDA) of the OTU level. OTUs were used as response data, and time point was explanatory data; the bacterial genera that contributed most are plotted. The covariance attributed to subject was first fitted by regression and then “partialled out” (removed) from the ordination. Variation explained by time point was 2.5%, *p* = 0.002. Ellipses cover 66% of the observations associated with each time point. Circles are *t* = 1 samples, and squares are *t* = 2 samples. Samples are separated based on time point on the first (horizontal) constrained axis, and the second (vertical) unconstrained axis captures a fraction of variation explained by factors other than time point.

**Table 1 nutrients-14-01204-t001:** Nutritional composition of placebo, WPC low and WPC high investigational products indicated as percentage by weight of the powder, except for calcium, which is expressed as mg/kg.

Description	Unit	Placebo	WPC Low	WPC High
Fat	*w/w*%	5	10	16
- of which Phospholipids	*w/w*%	0	4.4	7.0
Protein	*w/w*%	60	58	57
- of which HA300	*w/w*%	60	31	0
Lactose	*w/w*%	1.7	1.0	1.0
Maltodextrin	*w/w*%	18	18	18
Moisture	*w/w*%	4.4	4.7	4.0
Ash 525 °C	*w/w*%	7.7	5.3	4.0
Calcium content	mg/kg	7700	7967	8200

**Table 2 nutrients-14-01204-t002:** Baseline characteristics of subjects. No significant differences were observed between the groups with respect to age and BMI.

Variable		Placebo	Ingredient Low Dose	Ingredient High Dose
Gender	*n*	35	38	37
	Male	100%	100%	100%
Age	Mean (SD)	36.29 (11.6)	34.16 (11.93)	33.7 (9.98)
BMI (kg/m^2^)	Mean (SD)	24.43 (2.24)	23.85 (2.93)	24.02 (2.81)

**Table 3 nutrients-14-01204-t003:** Statistical study outcomes in relation to stool frequency.

Parameter	Variable	Placebo	WPC Low	WPC High	*p*-Value Trend
Change in stool frequency (# defecations/day) d16 vs. d15 ^P^	*n* (Nmiss)	35 (0)	38 (0)	37 (0)	*p* 0.01
	Mean (SD)	−0.9 (1.5)	−1.2 (1.6)	−0.4 (1.4)	
Percentage change in stool frequency d16 vs. d15 ^S^	*n* (Nmiss)	33 (2)	37 (1)	35 (2)	*p* 0.053
	Mean (SD)	−36.2 (45.2)	−38.8 (67.3)	−3.2 (80.8)	
Stool frequency d15 ^PH^	*n* (Nmiss)	35 (0)	38 (0)	37 (0)	*p* 0.931
	Mean (SD)	2.1 (1.4)	2.1 (1.4)	1.9 (1.1)	
Stool frequency d16 ^S^	*n* (Nmiss)	35 (0)	38 (0)	37 (0)	*p* 0.034
	Mean (SD)	1.2 (0.8)	1.0 (0.8)	1.5 (0.8)	
AUC for stool frequency d11-d18 ^PH^	*n* (Nmiss)	35 (0)	38 (0)	37 (0)	*p* 0.707
	Mean (SD)	9.2 (3.9)	8.5 (2.8)	8.9 (2.8)	

^P^ Primary, ^S^ secondary and ^PH^ post hoc analysis. PP analysis is shown. A significant trend between the 3 groups was observed for primary outcome change in stool frequency d16 vs. d15 but disappeared after correction for multiple testing (*p* > 0.025) according to the Hochberg procedure.

**Table 4 nutrients-14-01204-t004:** Statistical study outcomes in relation to diarrhea.

Parameter	Variable	Placebo	WPC Low	WPC High	Statistics
Change in GSRS diarrhea d16 vs. d15 ^P^	*n* (Nmiss)	35 (0)	38 (0)	37 (0)	*p* 0.13
	Mean (SD)	−2.8 (4.7)	−2.7 (4.6)	−1.4 (4.7)	
Percentage change in GSRS diarrhea d16 vs. d15 ^S^	*n* (Nmiss)	35 (0)	38 (0)	37 (0)	*p* 0.09
	Mean (SD)	−20.1 (37.9)	−21.3 (46.6)	−0.7 (61.0)	
GSRS max diarrhea score day 14–day 18 ^S^	*n* (Nmiss)	35 (0)	38 (0)	37 (0)	*p* 0.84
	Mean (SD)	8.3 (4.9)	7.4 (4.4)	7.9 (5.9)	
GSRS diarrhea d15 ^PH^	*n* (Nmiss)	35 (0)	38 (0)	37 (0)	*p* 0.48
	Mean (SD)	7.3 (4.9)	6.9 (4.5)	6.8 (5.7)	
AUC GSRS diarrhea d11-d18 ^PH^	*n* (Nmiss)	35 (0)	38 (0)	37 (0)	*p* 0.54
	Mean (SD)	29.7 (8.8)	29.2 (8.1)	30.4 (14.7)	

^P^ Primary, ^S^ secondary and ^PH^ post hoc analysis.

**Table 5 nutrients-14-01204-t005:** Stool consistency as measured by Bristol Stool Scale.

Parameter	Variable	Placebo	WPC Low	WPC High	Statistics
Change in BSS max d16 vs. d15	*n* (Nmiss)	28 (7)	26 (12)	34 (3)	*p* 0.79
	Mean (SD)	−0.82 (2.06)	−1.00 (1.7)	−0.38 (1.54)	
Percentage change in BSS max d16 vs. d15	*n* (Nmiss)	28 (7)	26 (12)	34 (3)	*p* 0.58
	Mean (SD)	−7.53 (43.01)	−14.88 (27.51)	−1.51 (36.11)	
BSS max d15	*n* (Nmiss)	33 (2)	37 (1)	35 (2)	*p* 0.07
	Mean (SD)	5.61 (1.60)	5.57 (1.48)	4.83 (1.64)	

BSS scores were calculated as absolute and percentage change in BSS at day 16 compared to day 15 and as maximum BSS score on d15. Parameters were all secondary outcomes, as defined in the statistical analysis plan.

## Data Availability

The data presented in this study are available on request from the corresponding author.

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
