# Peer review of "A Double-Blind, Randomized Intervention Study on the Effect of a Whey Protein Concentrate on E. coli-Induced Diarrhea in a Human Infection Model"

_nutrients, 2022, doi:10.3390/nu14061204_

Round 1
Reviewer 1 Report
The paper is good; however, the following comments still need to be further addressed. As shown below:
In the study, the authors present data suggesting that short term consumption of concentrated whey protein in healthy adults did not reduce the diarrhea score in the E. coli infection model and did not change the composition of intestinal microflora. They suggest that more research is needed to develop the potential of whey protein concentrate.
The following points should be addressed:
Introduction
- Insufficient whey protein concentrate background was provided to explain the necessity and innovation of the research.
- Could the strain represent coli-induced diarrhea? Is it universal?
- Is it reasonable that all subjects are male? Why not female?
- Is the dose and duration of coli sufficient to produce symptoms of diarrhea?
- In Figure 2, note is too large, resulting in overlap. Please change it. i.e. placebo (circles), WPC low (squares) and WPC high (triangles).
- Significant difference: the letter P should be italicized.
- What are the results of fecal SCFA analysis?
- I recommended to use some references in recent 10 years.
Author Response
We thank Reviewer 1 for his/he review and important remarks. Below, a point by point reply is given by the authors.
In the study, the authors present data suggesting that short term consumption of concentrated whey protein in healthy adults did not reduce the diarrhea score in the E. coli infection model and did not change the composition of intestinal microflora. They suggest that more research is needed to develop the potential of whey protein concentrate.
The following points should be addressed:
Introduction
Insufficient whey protein concentrate background was provided to explain the necessity and innovation of the research.
Thank you for this point. Whey protein concentrates are of interest for their potential as functional foods with specific anti-infection and immune supporting effects since they contain specific native whey proteins, mfgm protein and phospholipids. Therefore we studied the effect of this product in an infection model. We added a sentence to the introduction to make this more clear:
Line 66-68: “As described above, whey proteins and MFGM components such as specific proteins and phospholipids have been associated with anti-infection and immune related outcomes. The whey protein concentrate tested in this study contained these components.”
Could the strain represent coli-induced diarrhea? Is it universal?
The specific E. coli strain in our study is indeed used as a model (with obvious limitations) for a broader range of diarrheagenic E. coli strains. It is one of the many strains that have been used in human ETEC challenge studies for testing the efficacy of investigational drugs, vaccines and diet interventions (Porter et al, 2021). For the development of functional food ingredients, the more virulent strain models are considered less acceptable by ingredient manufacturers as well as ethical review boards. The challenge model using a live-attenuated E. coli strain (E1392/75-2A) induces mild, self-limiting diarrhea, as well as mild gastrointestinal symptoms, which do not require antibiotic treatment. The strain (tested as a vaccine) has been shown to confer 75% protection against re-infection with wild-type E. coli LT, ST, CFA/II strains, thus confirming that it represents wild-type diarrheagenic E. coli strains (Levine et al, 2019). We have reported the prevalence of WHO-defined diarrhea on the day after challenge with this strain and dose to be around 40% (Van Hoffen et al, 2020). Previous studies at NIZO food research showed that 38-53% of subjects reported diarrhea and that 38-40% experienced abdominal pain.
References added to the paper:
Levine, M. M., Barry, E. M. & Chen, W. H. A roadmap for enterotoxigenic Escherichia coli vaccine development based on volunteer challenge studies. Hum. Vaccin Immunother. 15, 1357–1378. https://doi.org/10.1080/21645515.2019.1578922 (2019).
Porter C.K., Talaat KR, Isidean SD, Kardinaal A, et al. (2021) The Controlled Human Infection Model for Enterotoxigenic Escherichia coli. In: Current Topics in Microbiology and Immunology. Springer, Berlin, Heidelberg. https://doi.org/10.1007/82_2021_242
Is it reasonable that all subjects are male? Why not female?
Women were not eligible for this study, because of the disadvantage of the female anatomy hindering fecal sample collection without urine contamination during acute infection, as well as interference of abdominal symptoms during their menstrual cycle.
Is the dose and duration of coli sufficient to produce symptoms of diarrhea?
Yes this is the case. Yet, it can be observed from Figure 2 that the variation between subjects is high, meaning that some score very high on the gastrointestinal symptom diarrhea scores (21) and some moderate (12) or do not have complaints (3). Since it was known that some would be non-responsive, this was taken along in the power calculations. As can be observed from Figure 2, the average on the symptoms of the total population on day 1 after infection is 7. This was on the subdomain diarrhea, which is typical an effect of E coli-induced infection. The diarrhea domain consist of 3 complaint items: a) diarrhea, a) loose stools, c) urgent bowel movement. A score of 1 means no complaints, and the maximum score is 7 and equals very severe complaints. So the minimum what is retrieved is a score of 3 (1 point of each of the three complaint items) and the maximum of 21 (7 points for each of the three complaint items).
Furthermore, as also mentioned above, we have previously reported the prevalence of WHO-defined diarrhea on the day after (single) challenge with this strain and dose to be around 40% (Van Hoffen et al, 2020). Previous studies at NIZO food research with the same dose showed that 38-53% of subjects reported diarrhea and that 38-40% experienced abdominal pain. In all E. coli challenge models, with various strains, there is a percentage of the study population that will not respond with diarrhea symptoms (Porter et al, 2021). In general, more virulent strains, used in vaccine development studies, aim at an “attack rate” of >60%.
In Figure 2, note is too large, resulting in overlap. Please change it. i.e. placebo (circles), WPC low (squares) and WPC high (triangles).
We adapted Figure 2 accordingly. Since the other reviewer requested color we adapted both color and the size of the notes.
Significant difference: the letter P should be italicized.
Thank you, noted. Changed throughout manuscript.
What are the results of fecal SCFA analysis?
We have now included information to the M&M section (Line 279), the results section (Line 395) and a bar graph of the SCFA results in the supplementary information Figure S3 and deleted “data not shown”.
I recommended to use some references in recent 10 years.
Thank you for this point, from the 54 references, 35 are of last 10 years (2012-now) and indeed almost 20 are from before. In dairy research some fields are only recently coming popular again since now is the time that industry can make on large scale specific milk fractions. The original findings, often based on lab or pilot scale ingredients, were made long ago (e.g in vitro effects of milkfat components on pathogens) that is why some references are very old. Indeed, these older studies are also summarized in a recent review of Brink et al (2020) that we cited (ref 8). Yet I do like to mention also the original articles that showed the findings that are relevant in the context of our manuscript. Hope you can agree on this.
Same accounts for material and methods section. We referred to quite some papers of which many are old, e.g. Bristol stool scale. Yet, these are the most appropriate to refer to so we would like to keep these. What we did do is add the paper of Levine of 2019 to the list (became ref 20) and the one of Porter (became ref 52 )(as discussed above).
Furthermore, some older reference is of lactoferrin and how similar it is to human LF in the original cloning paper. Again, this is the original one, referring to more recent with more in depth would not add to the paper.
Lastly, calculations about protein and how much his needed during infection are old. More recent papers on this topic are either out of scope (nutrition and covid) or too specific (protein requirements during critically illness).
In conclusion, we hope that you can accept our choice for references. If your have specific suggestions for articles we may have missed we are of course open to this.
Reviewer 2 Report
In this Randomized Intervention Study, Ulfman et al. find that the dietary intake of whey protien concentrate does not change the outcome of diarrheagenic E. Coli challenge and does not change the composition of microbiota. Despite the study reports negative results it is well-crafted and scientifically rigorous. The only flaw of the study that it somewhat squanders the potential and data in the microbiome analysis that needs more detailed elaboration. Therefore, the Reviewer recommends the following revisions before publication:
Major Remarks:
The microbiota analysis is not profound enough even for a 16S RNA analysis and the Authors' conclusion that no changes are detected in neither experimental groups are somewhat superficial. While the results for the primary outcome clearly indicate the validity of the negative results, that is also impactful for its scientific field, further work needs to be done for microbiota analysis.
- The Authors should add at least one more diversity (Shannon, Simpson, etc) index to Faith 's PD and calculate richness, though it is highly relevant in assessing microbiome in the context of diet change.
- Line 377-379 "Also, no association of microbiota composition with complaint scores was found (data not shown). In line with the lack of difference in microbiota composition between the intervention groups, also no difference in fecal SCFA composition was observed (data not shown)" --> please show the data (at least in supplemental). The Authors are reporting negative results, data availability is especially important.
- An interesting side-finding of the study is that follow-up microbiota composition changes relative to baseline samples in all groups and this may be because the dietary restriction of dairy products. Though the Authors discuss that other studies already got similar results, still in this setting it is worth to elaborate a bit more on this in the Results section (Also, please visualize the longitudinal changes in key genera described in the results at least in a supplemental figure).
Minor Remarks:
- Please color Figure 2 for better presentation
- Figure 3 on RDA needs to be redrawn (please arrange taxa labels to be legible)
- Please describe the meaning of Axes in figure legends of Figure 3
- Please add to the limitation section that the trial did not evaluate the effect of different doses of whey protein intake. 30 g is not a big dose anyway. Would a higher dose have considerable effect on the primary and secondary outcomes, or microbiota composition?
Author Response
We thank Reviewer 2 for his/he review and important remarks. Below, a point by point reply is given by the authors.
Reviewer 2:
In this Randomized Intervention Study, Ulfman et al. find that the dietary intake of whey protein concentrate does not change the outcome of diarrheagenic E. Coli challenge and does not change the composition of microbiota. Despite the study reports negative results it is well-crafted and scientifically rigorous. The only flaw of the study that it somewhat squanders the potential and data in the microbiome analysis that needs more detailed elaboration. Therefore, the Reviewer recommends the following revisions before publication:
Major Remarks:
The microbiota analysis is not profound enough even for a 16S RNA analysis and the Authors' conclusion that no changes are detected in neither experimental groups are somewhat superficial. While the results for the primary outcome clearly indicate the validity of the negative results, that is also impactful for its scientific field, further work needs to be done for microbiota analysis.
The Authors should add at least one more diversity (Shannon, Simpson, etc) index to Faith 's PD and calculate richness, though it is highly relevant in assessing microbiome in the context of diet change.
Thank you for giving us the opportunity to add more data. We have now incorporated findings from the Shannon index and richness (Line 377-381, line 481,491 and Supplementary Figure S1). There was no effect of treatment on any of these metrics, but comparing time points, there was a statistically significant increase in Shannon diversity (not for PD and richness).
Line 377-379 "Also, no association of microbiota composition with complaint scores was found (data not shown). In line with the lack of difference in microbiota composition between the intervention groups, also no difference in fecal SCFA composition was observed (data not shown)" --> please show the data (at least in supplemental). The Authors are reporting negative results, data availability is especially important.
We agree with Reviewer 2 that data availability is important. We’ve now included information to the M&M section (Line 279), the results section (Line 395) and a bar graph of the SCFA results in the supplementary information Figure S3.
An interesting side-finding of the study is that follow-up microbiota composition changes relative to baseline samples in all groups and this may be because the dietary restriction of dairy products. Though the Authors discuss that other studies already got similar results, still in this setting it is worth to elaborate a bit more on this in the Results section (Also, please visualize the longitudinal changes in key genera described in the results at least in a supplemental figure).
We agree it is worth to elaborate a bit more. In the result section (Line 387-390) we now refer to a new supplementary table (S2) in which we summarized the taxa that were significantly different between the two time points, after correction for multiple testing (M&M section (line 290-291)).
Minor Remarks:
Please color Figure 2 for better presentation
We adapted Figure 2 accordingly. Since the other reviewer requested the size of the notes we adapted both colour and the size of the notes.
Figure 3 on RDA needs to be redrawn (please arrange taxa labels to be legible)
We have redrawn figure 3 and replaced the old figure 3 in the manuscript.
Please describe the meaning of Axes in figure legends of Figure 3
We’ve now added this to the figure legend of Figure 3.
Please add to the limitation section that the trial did not evaluate the effect of different doses of whey protein intake. 30 g is not a big dose anyway. Would a higher dose have considerable effect on the primary and secondary outcomes, or microbiota composition?
We added this to the limitation section (line 455-457).
It is very difficult to estimate whether a higher dose would have had an effect. Yet, we were disappointed that this dose did not have an effect since a previous study (Ten Bruggencate et al, 2016) used slightly lower amount of product and did see an effect in a similar mode
Round 2
Reviewer 2 Report
The Authors have sufficiently provided all the details required by the Reviewer and improved the manuscript noteworthy. All issues have been successfully assessed.